

# CircParser: a novel streamlined pipeline for circular RNA structure and host gene prediction in non-model organisms

Artem Nedoluzhko[1,*], Fedor Sharko[2,3,*], Md. Golam Rbbani[1], Anton Teslyuk[2], Ioannis Konstantinidis[1] and Jorge M.O. Fernandes[1]

[1] Faculty of Biosciences and Aquaculture, Nord University, Bodø, Bodø, Norway
[2] Complex of NBICS Technologies, National Research Centre "Kurchatov Institute", Moscow, Russia
[3] Institute of Bioengineering, Research Center of Biotechnology of the Russian Academy of Sciences, Moscow, Russia, Russia
[*] These authors contributed equally to this work.

Corresponding authors
Artem Nedoluzhko,
artem.nedoluzhko@nord.no
Jorge M.O. Fernandes,
jorge.m.fernandes@nord.no

## ABSTRACT

Circular RNAs (circRNAs) are long noncoding RNAs that play a significant role in various biological processes, including embryonic development and stress responses. These regulatory molecules can modulate microRNA activity and are involved in different molecular pathways as indirect regulators of gene expression. Thousands of circRNAs have been described in diverse taxa due to the recent advances in high throughput sequencing technologies, which led to a huge variety of total RNA sequencing being publicly available. A number of circRNA de novo and host gene prediction tools are available to date, but their ability to accurately predict circRNA host genes is limited in the case of low-quality genome assemblies or annotations. Here, we present CircParser, a simple and fast Unix/Linux pipeline that uses the outputs from the most common circular RNAs *in silico* prediction tools (CIRI, CIRI2, CircExplorer2, find_circ, and circFinder) to annotate circular RNAs, assigning presumptive host genes from local or public databases such as National Center for Biotechnology Information (NCBI). Also, this pipeline can discriminate circular RNAs based on their structural components (exonic, intronic, exon-intronic or intergenic) using a genome annotation file.

## INTRODUCTION

De novo genome sequencing has become a routine procedure, due to a decrease in sequencing costs, diversification of high-throughput sequencing platforms and improvement of bioinformatic tools (*Ekblom & Wolf, 2014*). However, the quality of non-model species genome assemblies and, as a result, their annotations are often of unsatisfactory quality, because of (1) repetitive sequences, including transposons, and short sequence repeats (SSRs); (2) gene and genome duplications; (3) single-nucleotide polymorphisms (SNPs) and genome rearrangements (*Lien et al., 2016*; *Negrisolo et al., 2010*; *Rodriguez & Arkhipova, 2018*; *Yahav & Privman, 2019*).

CircRNAs are relatively poorly studied members of the non-coding RNA family. These unique single-stranded molecules are generated through back-splicing of pre-mRNAs in a wide range of eukaryotic and prokaryotic taxa (*Danan et al., 2012*; *Holdt, Kohlmaier & Teupser, 2018*), and even viruses (*Huang et al., 2019*). CircRNAs play a significant role in the regulation of the molecular pathways not only through modulating of microRNA and protein activity, but also by the affecting transcription or splicing (*Holdt, Kohlmaier & Teupser, 2018*).

These regulatory molecules have been known for decades, but the development of high-throughput DNA analysis methods lead to a rapid increase in the number of studies related to these type of non-coding RNAs. This, in turn, resulted in a requirement for additional circRNA prediction tools. The miARma-Seq (*Andres-Leon & Rojas, 2019*) with CIRI predictor (*Gao, Wang & Zhao, 2015*), circRNA_finder (*Westholm et al., 2014*), find_circ (*Memczak et al., 2013*), CIRCexplorer2 (*Zhang et al., 2016*), and other tools are very popular today for prediction of circRNAs sequences based on transcriptomic data (*Hansen et al., 2016*; *Szabo & Salzman, 2016*), despite significant output differences. Several circRNA predictors (CIRI, CIRI2, and CircExplorer2) can use genome annotation files for host gene prediction but they are definitely useful only for well-annotated genomes, and even, such as CircView (*Feng et al., 2018*) or circMeta (*Chen et al., 2019*), have been designed specifically for them.

Here we describe CircParser, a novel and easy to use Unix/Linux pipeline for prediction of host gene circular RNAs using the blastn program and the freely available bedtools software (*Quinlan & Hall, 2010*). CircParser can be also implemented as a part of pipelines for *de novo* prediction of circular RNA because of its versatile output files. CircParser is most useful for circRNA host gene prediction analysis in whole transcriptomic datasets for low-quality assembled, as well as poorly annotated genomes. It sorts and joins overlapped circular RNAs sequences and predicts host gene name for overrepresented circRNAs, while identifying their structural components. We demonstrate the prediction capacity of CircParser on a recently published transcriptomic data set from the wild and domesticated females of Nile tilapia (*Oreochromis niloticus*) fast muscle (Konstantinidis et al., under review) using the five most popular circRNAs in silico prediction tools—CIRI, CIRI2, CircExplorer2, find_circ, and circFinder.

## MATERIALS & METHODS

The results of Illumina sequencing of twelve ribosomal RNA depleted RNA-seq libraries reads have been downloaded from Gene Expression Omnibus (accession number GSE135811). The DNA reads were filtered by quality (phred > 20) and library adapters were trimmed using Cutadapt software (version 1.12) (*Marcel, 2011*). The Nile tilapia reference genome (ASM185804v2) and its gene-annotation (ref_O_niloticus_UMD_NMBU_top_level.gff3) were used in the following analysis.

CircRNA prediction was performed for each ribosomal RNA depleted RNA-seq library using the circRNA in silico prediction tools (i) CIRI (*Gao, Wang & Zhao, 2015*) that is linked to miARma-Seq pipeline (*Andres-Leon & Rojas, 2019*), (ii) CIRI2 (*Gao, Zhang

**Table 1  CircParser.pl usage. Required and optional parameters.**

| Parameter | Parameter description |
| --- | --- |
| -h, –help | Show this help message and exit |
| -b | CircRNA input file (required) |
| -g, –genome | Reference genome file (required) |
| -t, –tax | NCBI TaxID (optional) |
| -a | Genome annotation file, gff/gff3 file (optional) |
| –np | Prohibition for coordinate merging (optional) |
| -c, –ciri | Input circRNA from CIRI\|CIRI2 *in silico* predictors, (default: input from CircExplorer2, find_circ, circFinder, and BED files) |
| –threads | Number of threads (CPUs) for BLAST search (optional) |
| -v, –version | Current CircParser version |

& Zhao, 2018), (iii) CircExplorer2 (*Zhang et al., 2016*), (iv) find_circ (*Memczak et al., 2013*), and (v) circFinder (*Westholm et al., 2014*). Prediction output files from all libraries were converted separately to coordinate file format. After sorting, these coordinate files (from different prediction algorithms, but for each library) were merged using bedtools multiinter (*Quinlan & Hall, 2010*) to determine a joint prediction output from CIRI, CIRI2, CircExplorer2, find_circ, and circFinder (see Table S1).

We developed CircParser, as a streamlined pipeline, which makes use output files from the most popular circRNAs in silico predictors. CircParser works under Linux/Unix system and its parameters are presented in Table 1.

*Usage: perl CircParser.pl [-h] -b INPUT_FILE—genome REF_GENOME*

CircParser can merge overlapped circRNAs coordinates from circRNAs predictor outputs using bedtools merge (*Quinlan & Hall, 2010*) at the first stage of the pipeline; this ensures that they are related to the same host gene and creates separate coordinates files (bed file) with overlapped circRNAs coordinates. In addition, it is optionally possible to merge circRNA without overlapping coordinates but located in the contiguous genome locus using the special option.

The separate coordinate files (bed file) are converted to fasta files using bedtools getfasta (*Quinlan & Hall, 2010*). Finally, CircParser uses fasta files for host gene prediction using a NCBI database (the longest stage of pipeline) for circRNAs (Fig. 1A). CircParser works by default with the NCBI online database, but it can optionally use a custom database or a pre-compiled NCBI database installed locally. CircParser includes the following blast parameters, which are necessary for host gene prediction, and assigns sequences to the respective circRNA: *-perc_identity* 90; *-max_target_seqs* 1000; *-max_hsps* 1; the maximum number of aligned sequences to keep is 1000; the minimum percent identity of matches to report is 90%. CircParser also filters out non-informative blast results, such as "uncharacterized", "clone", "linkage group" and others from the output table.

CircParser can also discriminate circular RNAs by their structural components: exonic, intronic, exon-intronic or intergenic using genome annotation gff/gff3 file (-a parameter). In this case, the user should avoid circRNAs coordinate merging (using –np parameter) during the pipeline implementation for correct results (Fig. 1B).

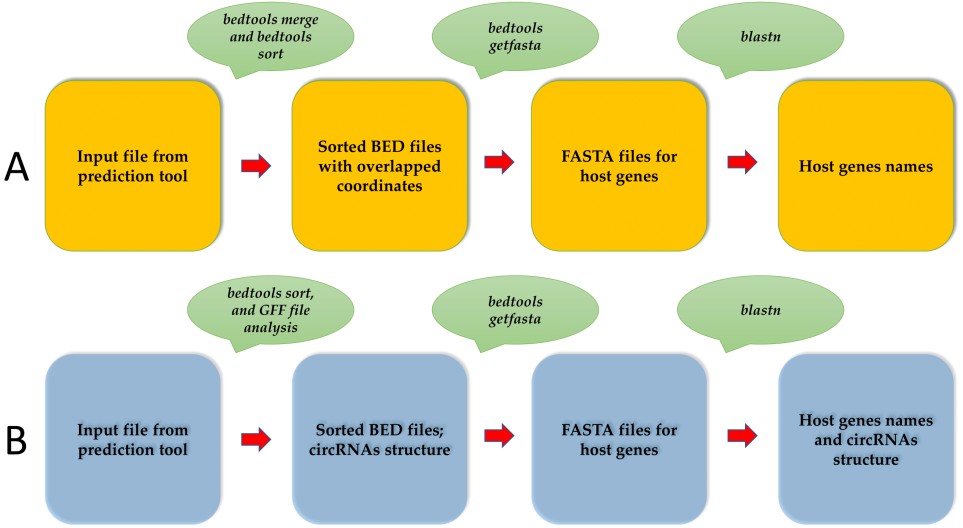

**Figure 1 An overview of the CircParser pipeline.** (A) The pipeline includes merging of the circRNAs with overlapping genome coordinates and presents the number of different circRNAs originating from one host gene. (B) CircParser includes the prediction of circRNA structural components using a genome annotation gff/gff3 file.

*Usage: perl CircParser.pl -np -b INPUT_FILE –genome REF_GENOME -a GENOME.gff*

However, poor quality of annotation file can lead to errors in the circRNAs structure analysis.

The Perl implementation of CircParser is available at https://github.com/SharkoTools/CircParser.

## RESULTS

We applied CircParser to twelve merged coordinate files that contained information about joint coordinates for circRNAs predicted using CircExplorer2, miARma-Seq (with CIRI predictor), CIRI2, find_circ, and circFinder. The five different algorithms predicted on average ∼131 (CircExplorer2); ∼501 (CIRI); ∼706 (CIRI2); ∼257 (find_circ), and ∼398 (circFinder) circRNAs per sample, with an insignificant overlap ∼37 circRNAs (Fig. 2; Table S1), similarly to previously published comparisons (*Hansen, 2018*; *Hansen et al., 2016*).

To access the host gene of circular RNAs and to reduce false-positive rates, only overlapping circRNAs (Fig. 2) were used in CircParser. This pipeline allows the elimination of non-informative outputs (e.g., contains only chromosome/contig name, number of uncharacterized loci, or name of BAC clone, and etc.), while keeping more the relevant blast results and retrieving the likely host gene name for the circular RNAs; in the case of impossibility to find identical sequences in the database, this tool mark these sequences as NOT ASSIGNED).

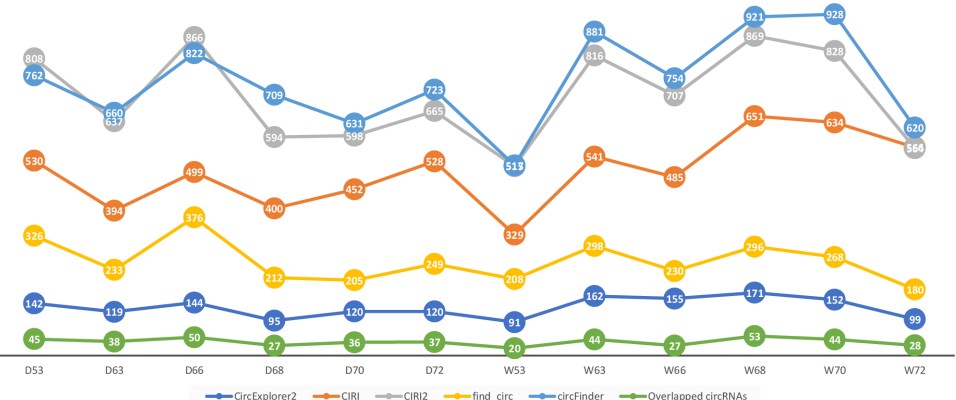

**Figure 2   Number of circular RNAs that have been predicted by CIRI, CIRI2, CircExplorer2, find_circ, circFinder, and that are common between all prediction algorithms.**

## DISCUSSION

The CircParser results also allow us to determine the number of circRNA types from one host gene and their minimum and maximum size in base pairs (bp). We showed that our algorithm detected presumable host gene names for the vast majority of predicted circRNAs. Moreover, most of them were related to muscle functions (e.g., *calcium/calmodulin-dependent protein kinase*, *troponin T3*, *myocyte-specific enhancer factor 2C*, and others), and immune-related genes (*MHC class IA antigen*), which were consistently found among different individuals (Table S2), despite the relatively low coverage for circRNAs analysis of the sequencing data used (*Mahmoudi & Cairns, 2019*). An example of circRNA structure analysis for CIRI, CIRI2, CircExplorer2, find_circ, and circFinder outputs is presented in Supplementary Table S3.

To estimate the capacity of our pipeline we compared a number of host genes that were predicted by CircExplorer2 and CircParser (CircExplorer2 outputs were used as input files) for the same *O. niloticus* fast muscle datasets used earlier. As a result, CircParser shows greater efficiency for Nile tilapia, improving the number of predicted host genes up to two-fold (Fig. 3).

Another equally important aspect of CircParser concerned the accuracy of this pipeline. The most well-annotated reference genome of zebrafish (assembly GRCz11) and zebrafish muscle transcriptomic dataset (ERR145655) were used for accuracy estimation, i.e., the agreement between the annotation file and CircParser output. We showed that in this case, CircParser host gene prediction was confirmed in 82.4% cases.

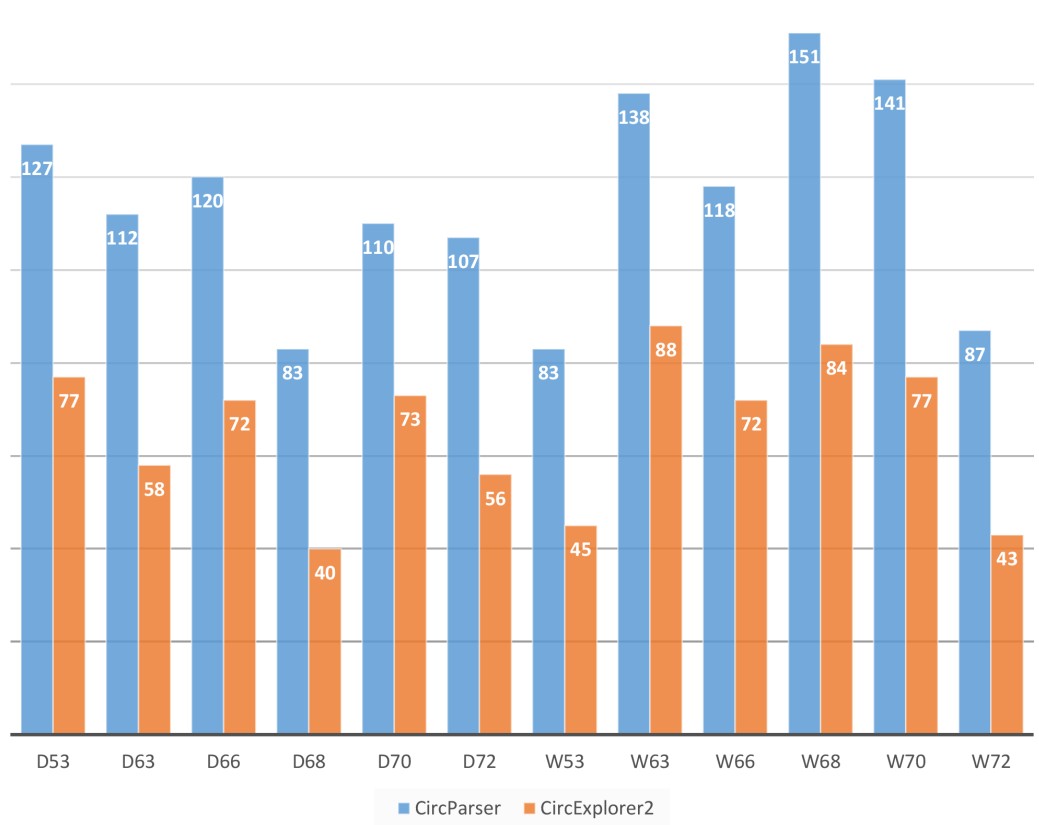

**The number of predicted host genes for Nile tilapia datasets**

**Figure 3** CircParser capacity: number of host genes that were predicted by CircExplorer2 and Circ-Parser.

## CONCLUSIONS

Thus, we conclude that CircParser represents a reproducible workflow that enables researchers to effectively predict the host genes for circular RNAs, even in non-model organisms with poorly annotated genome assemblies.

## ACKNOWLEDGEMENTS

We would like to acknowledge Jorge Galindo-Villegas from Nord University (Norway) and Tomas B. Hansen from the Aarhus University (Denmark) for their valuable advice.

### Funding

This study has received funding from the European Research Council (ERC) under the European Union's Horizon 2020 research and innovation programme (grant agreement no 683210) and from the Research Council of Norway under the Toppforsk programme (grant agreement no 250548/F20). Fedor Sharko was supported by the RFBR (Russian Foundation for Basic Research) Grant no 19-54-54004. This work was partially carried out in Kurchatov Center for Genome Research and supported by Ministry of Science and Higher Education of Russian Federation, grant #075-15-2019-1659. The funders had no role in study design, data collection and analysis, decision to publish, or preparation of the manuscript.

### Grant Disclosures

The following grant information was disclosed by the authors:
European Research Council (ERC) Consolidator Grant: 683210.
Research Council of Norway under the Toppforsk programme: 250548/F20.
RFBR (Russian Foundation for Basic Research): 19-54-54004.
Ministry of Science and Higher Education of Russian Federation: #075-15-2019-1659.

### Competing Interests

Jorge M.O. Fernandes is an Academic Editor for PeerJ. The authors declare there are no competing interests.

### Author Contributions

- Artem Nedoluzhko conceived and designed the experiments, performed the experiments, analyzed the data, prepared figures and/or tables, authored or reviewed drafts of the paper, and approved the final draft.
- Fedor Sharko conceived and designed the experiments, analyzed the data, prepared figures and/or tables, and approved the final draft.
- Md. Golam Rbbani analyzed the data, authored or reviewed drafts of the paper, and approved the final draft.
- Anton Teslyuk analyzed the data, prepared figures and/or tables, authored or reviewed drafts of the paper, and approved the final draft.
- Ioannis Konstantinidis performed the experiments, authored or reviewed drafts of the paper, and approved the final draft.
- Jorge M.O. Fernandes conceived and designed the experiments, authored or reviewed drafts of the paper, and approved the final draft.

### DNA Deposition

The following information was supplied regarding the deposition of DNA sequences:
Data is available at the NCBI Gene Expression Omnibus database: GSE135811.

## Data Availability

CircParser is available at Github. Availability and implementation: The Perl implementation of CircParser is available at https://github.com/SharkoTools/CircParser.

## Supplemental Information

Supplemental information for this article can be found online at http://dx.doi.org/10.7717/peerj.8757#supplemental-information.

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
