# Peer review of "CircParser: a novel streamlined pipeline for circular RNA structure and host gene prediction in non-model organisms"

_PeerJ, doi:10.7717/peerj.8757_

## Round 0.1 · original submission · Major Revisions

The reviewer had critical remarks regarding accuracy and citing recent literature (please look suggestions by reviewer #1). Please add some discussion about advantage of the software developed in comparison to other tools. I believe this is novel application of interest for biologists studying non-model organisms. Encourage you revise and resubmit it soon, by 2019.

Reviewer 1 ·

Basic reporting

The article was written in comprehensive English, was well structured, and was easy to understand. The references were appropriate but the authors should have a look at DOI:10.1038/ncomms12060 describing alternative splicing in circRNAs and DOI:10.1093/bioinformatics/btv656 describing a more accurate circRNA detection tool than CIRI.

This is a methods paper with a proof of principal application so it is lacking any major findings. Furthermore, the authors fail to show the information gained by using their program. I suggest two control experiments:
1. of the dataset the used, please show how many circRNAs were annotated with host-genes by the individual circRNA prediction tool, i.e. how many circRNAs were assigned to genes if circRNA detection was run with a reference gtf. How many more circRNAs can be assigned to genes using their program CircParser.
2. To show the overall accuracy, I suggest to use a dataset of a well annotated organism such as human or mouse. Show how many circRNAs are correctly associated with their host-gene.

Minor Remarks.
There were some formatting issues in line 51.
There was a spelling mistake of a program in line 93 it should be "bedtools multiinter"
The resolution of the figures is too small.
Figure legend 1b is not informative.
The raw data is only accessible after August 1st 2020.

Experimental design

The article would fit in the scope of PeerJ.
The research statement was well defined.
The authors should consider running the in silico experiments described above to really show the advantage of their program and make the paper strong and convincing. Currently I am not convinced that I would gain anything by running their program. I would also appreciate it, if the authors showed results for using only one circRNA prediction tool at a time as this would be the more used case.
As this is a methods paper, the authors need to describe the methods in more detail. Especially how they select the appropriate blastn result and assign it to the respective circRNA.
The provided git repository should include a use-case with input, command line and output.

Validity of the findings

In Line150 the authors conclude that their program "represents a fast and reproducible workflow", however they show neither any run-time analyis, nor any results showing reproducibility.

Additional comments

The paper provides a very nice idea and implementation on how to predict circRNA host genes in non-model organisms, however I would highly appreciate if the authors could include the suggestions mentioned above in order to show the validity and information gain of their program.

·

Basic reporting

no comment

Experimental design

no comment

Validity of the findings

no comment

Additional comments

Dear authors, it was nice article to read and learn about, however I have some minor remarks that should improve your article.
1. As I understood, the main benefit of CircParser among other tools is that it uses converted to fasta sequences and blast to search for host genes. So I think that you should add more about this benefit in article, because now it's not so clear.
2. Pipeline works only in Unix environment, why don't to put it inside Docker image, as well as other tools and provide single docker-compose file that will do all the analyses in any OS (Linux, Windows, MacOS).

---

## Round 0.2 · accepted · Accept

Thank you for the manuscript updates. The reviewers have no more remarks.

Reviewer 1 ·

Basic reporting

no comment

Experimental design

no comment

Validity of the findings

no comment

Additional comments

Thank you for addressing my main concerns.

·

Basic reporting

no comment

Experimental design

no comment

Validity of the findings

no comment

Additional comments

Text was improved and now it's easier to follow general idea of article. Hope to see dockerized version of this software in near future.